# Determination of the Porosity Distribution during an Erosion Test Using a Coaxial Line Cell

**DOI:** 10.3390/s19030611

**Published:** 2019-02-01

**Authors:** Tilman Bittner, Mathieu Bajodek, Thierry Bore, Eric Vourc’h, Alexander Scheuermann

**Affiliations:** 1School of Civil Engineering, University of Queensland, St Lucia, QLD 4072, Australia; mathieu.bajodek@ens-cachan.fr (M.B.); t.bore@uq.edu.au (T.B.); 2Laboratory of Systems and Applications of Information and Energy Technologies (SATIE UMR8029), ENS Paris Saclay, 94230 Cachan, France; Eric.Vourch@satie.ens-cachan.fr (E.V.); a.scheuermann@uq.edu.au (A.S.)

**Keywords:** time domain reflectometry, porosity measurements, inversion, sensor validation, internal erosion

## Abstract

The detection of porosity changes within a soil matrix caused by internal erosion is beneficial for a better understanding of the mechanisms that induce and maintain the erosion process. In this paper, an electromagnetic approach using Spatial Time Domain Reflectometry (STDR) and a transmission line model is proposed for this purpose. An original experimental setup consisting of a coaxial cell which acts as an electromagnetic waveguide was developed. It is connected to a transmitter/receiver device both measuring the transmitted and corresponding reflected electromagnetic pulses at the cell entrance. A gradient optimization method based on a computational model for simulating the wave propagation in a transmission line is applied in order to reconstruct the spatial distribution of the soil dielectric permittivity along the cell based on the measured signals and an inversion algorithm. The spatial distribution of the soil porosity is deduced from the dielectric permittivity profile by physically based mixing rules. Experiments were carried out with glass bead mixtures of known dielectric permittivity profiles and subsequently known spatial porosity distributions to validate and to optimize both, the proposed computational model and the inversion algorithm. Erosion experiments were carried out and porosity profiles determined with satisfying spatial resolution were obtained. The RMSE between measured and physically determined porosities varied among less than 3% to 6%. The measurement rate is sufficient to be able to capture the transient process of erosion in the experiments presented here.

## 1. Introduction

The main characteristic of internal erosion is the transient dislodgement and transport of fine particles in the soil matrix or at interfaces of different soil layers causing changes in porosity and density. However, the mechanisms that induce and maintain the process of internal erosion are not yet clearly understood [1].

A better understanding can be achieved with an integrated approach of both the grains and pores at the micro scale as well as of the entire soil layers at the macro scale. It was lately shown that the shapes of pores and pore channels at the micro scale have a significant influence on the local flow conditions [1,2,3]. These parameters, which are characteristic on the micro scale, are quantified by the porosity distribution at the macro scale and are interdependent on the mechanical and hydraulic properties. For this reason, the monitoring of the spatial porosity distribution and its alteration during erosion experiments is helpful to improve our understanding of this complex process.

Different approaches have been reported in the literature for this purpose of porosity measurements in a wide range of experiments and tests. One possibility is the determination of the rate of the washed out fine fraction [4,5]. The weight of this fraction collected downstream of the soil sample was recorded for a given time period and hydraulic boundary condition. This was partly accompanied by sieve analysis of the washed out fraction as well as the remaining coarse soil skeleton. As a result, conclusions can be drawn about the critical grain size distribution and the hydraulic boundary conditions. However, this method is limited to specific setups, for example suffusion tests with a downwards flow and does not allow observations about transient alterations in spatial porosity distributions during the test. Conclusions can be drawn out of visual observations in changes of layer heights during tests and layer wise sampling afterwards [6]. Particle image velocimetry (PIV) can be used for tracing the transport of fine particles and changes of the coarse matrix but it is limited to surface layers [7]. 

Detailed 2D and 3D pictures of soil specimens can be made with computed tomography scans (CT), enabling the localisation of individual particles and pores with a high resolution in order to allow analyses of internal stability and erodibility of fine fractions [8,9]. A drawback is the complex sample preparation procedure [8], which makes scans during tests impossible. Furthermore, the resolution is linked to the sample size; a detailed scan is only possible for small samples [10].

Porosity profiles of a sample before, during and after a test were determined by means of a gamma ray source and a scintillation counter [11,12]. With this technique, the sample is scanned layer wise. There is a time delay between the scan of the first and last layer, which may lead to measurement deviations in fast changing conditions. Additionally, occupational health and safety considerations need to be considered to reduce the risk of radiation. 

Another possibility consists in measurement techniques based on the interaction of electromagnetic fields with the surrounding material. Such approaches take advantage of the dipolar character of water molecules resulting in high electric permittivity in comparison to other phases such as solid and gas. This permittivity contrast between the different phases can be used to determine soil properties by implementing high frequency (in the MHz to GHz range) electromagnetic methods in time domain [13] or in frequency domain [14]. By far, TDR (Time Domain Reflectometry) is the most popular measurement technique for the water content of soils and has been established since the 1970s [15,16,17,18].

A further development allowed the additional measurements of the soils density [19]. Only a local or mean value can be determined with conventional TDR [13]. With the development of an efficient inversion algorithm [20,21], the computation of water content profiles along probes of several meter length became possible. For a better distinction, the term STDR (Spatial TDR) was created for his method. 

The latter approach has been successfully applied for the measurement of water content distributions in river dikes [22] as well as the determination of porosity profiles of samples with a rod probe [23]. STDR is a rather cost-effective method that allows for measurements featuring a spatial resolution in the order of centimetres. When combined with a fast inversion process, such a technique can provide nearly real time measurements [22].

An important requirement to perform STDR is a suitable sensor design. The sensor in form of a transmission line is inserted in the material to be characterized. The electromagnetic pulse travels along the transmission line and the transmitted and/or the reflected signals are recorded. The design of the sensor is fundamental, since it has to fulfil several important criteria that also involves the material under test (MUT). Therefore, one criteria concerns the representative elementary volume (REV) with respect to the allowable maximum particle size. According to Robert [24], material sample dimensions need to be at least three times greater than the maximum dimensions of the major aggregate. Hence, a transmission line with a larger sensitive area is required for coarse grained materials. Another criterion relates to the cohesiveness and viscosity of the soil. Both are influencing the installation of sensors.

Fork type probes are often used to determine the moisture of soils, which is the main application of dielectric methods in geotechnical engineering and soil science [19]. These probes can easily be pushed into a soil, which means easy handling for laboratory and field application. However, such instrumentation systems are often built only to deliver an average or a local value, as the length of the rods is usually limited. 

Further developments lead to flat ribbon cables (e.g., as a three-wire transmission line embedded in a polyethylene insulation) for measuring sections with a length up to several meters. A drawback of this type of sensor is that the electromagnetic field distribution depends on the dielectric permittivity of the surrounding media and is therefore not homogenous along the sensor. 

To avoid this drawback, a coaxial arrangement can be used. In this configuration, the electromagnetic field is more specified since it is concentrated between the inner and outer conductor. Coaxial arrangements have been commonly used for the dielectric characterisation of liquids [25] and soils [26,27]. The soil samples are either prefabricated or prepared directly in the cell. Hence the change of a state parameter of the samples, such as the water content or the density, during the test is usually not possible and mostly also not intended.

In this paper, we propose a newly designed experimental setup featuring a coaxial arrangement ideal for electromagnetic measurement techniques with a view to run an erosion test within the coaxial sensor. In the following, Section 2 describes the experimental setup implementing the coaxial erosion cell which is filled with glass beads in fully water-saturated conditions so as to enable focusing on porosity changes during the erosion process. In Section 3, the functional principle of Spatial TDR is described, which first consists inverting the TDR-signal into a capacitance profile of the cell, second to determine from the latter to a permittivity profile and finally to determine a porosity profile of the MUT. Section 4 reports on the validation of the inversion process thanks to measurements under known boundary conditions, which means under known porosity profiles. Finally, Section 5 provides the first results of porosity changes during erosion tests.

## 2. Experimental Setup

### 2.1. General Setup

The test setup is from the hydraulic point of view developed based on a constant head permeability test (Figure 1). The hydraulic boundary conditions can be adjusted by means of a moveable constant head overflow tank upstream of the sample, whereas the hydraulic potential downstream is fixed by a constant overflow at the top of the erosion cell. Adjustments of the flow rate for a given hydraulic head are not possible. The water circulates in a closed system, consisting of a 100 litres reservoir with a submerged pump, a control valve, the constant head overflow, the erosion cell with the downstream overflow and a flow meter. To achieve negligible head losses in the pipework for the flow rates occurred in the tests, all connecting pipes and hoses have an inner diameter of 32 mm. For the same reason, the flow meter is mounted downstream of the erosion cell, as its smaller inner diameter acts as a flow constriction. 

The hydraulic conditions in the cell, namely the pore water pressures along the soil column, are monitored with 14 pressure transmitters WIKA A-10 (WIKA Alexander Wiegand SE & Co. KG, Klingenberg, Germany) with an accuracy of 0.5% according to the manufacturer’s data sheet. The transmitters are arranged on the side wall of the cell in vertical distances between 25 mm and 50 mm. The closer interval is set where the main alterations during the erosion tests are expected. Flow rate measurements are conducted by means of a displacement flowmeter ManuFlo MES20-S-T (Manu Electronics Pty Ltd, Sydney, Australia) at an accuracy of 1.5% as stated by the manufacturer. Both, hydraulic head and flow rate, are automatically recorded at an interval of 10 s with a data logger. With these two parameters, the hydraulic gradient and the hydraulic conductivity of the soil can be determined.

### 2.2. Coaxial Cell

The coaxial erosion cell aims at characterising the changes of state of the MUT placed inside the cell throughout the measurements by quantifying the porosity distribution. Such changes are induced by a controlled water flow. Hence, the experimental set-up is optimised both to fulfil the required hydraulic boundary conditions and to perform the electromagnetic measurements.

The cell is manufactured from commercially available coaxial rigid transmission line components manufactured by the company Spinner GmbH. The use of this components is beneficial in regards of the fabrication costs of the cell. In particular, the cell implements a readily available conical transition for connecting the cylindrical core to a coaxial cable connected to the TDR device which both feeds the cell with the electromagnetic signals and allows measurements of the reflected signal. The drawback is that only a limited range of dimensions is commercially available. The main features of the tubes selected for the cell are provided in Table 1.

Fittings and connections for the water supply pipe and the pressure transducers required only comparatively small holes drilled in the outer tube, which does not disturb the electromagnetic field. In addition, a distance piece and flow homogeniser is used to evenly distribute the water flow at the inflow to the soil sample. This piece is a perforated plate made of 10 mm thick PMMA (Poly(methyl methacrylate), acrylic glass). The perforated plate homogenizes the inflow of water, leading to a more uniform boundary condition at the upstream side of the sample. With the low dielectric permittivity of the PMMA, which is about 3.7, the measured TDR-signal features a specific peak before entering the sample, which is useful for the analysis of the signal as it marks the beginning of the soil column. 

A main purpose of the cell is to induce an erosion process through a water flow and to observe the accompanying porosity changes with Spatial TDR. Moreover, the possibility of observing the particle movements in the cell is important for a better understanding of the erosion process as well as a kind of control for the TDR analysis. For this purpose, a 420 mm high and 40 mm wide inspection window was included in the cell (Figure 1 and Figure 2). Given the 155.6 mm outer diameter of the tube, the width of the window represents approximately 8.2% of the circumference. According to electromagnetic numerical simulations based on finite element modelling, the influence on the electromagnetic field distribution of such a relatively small opening in the outer tube is negligible.

In order to achieve a larger annulus to provide more space for the soil particles, an inner conductor with a smaller diameter than necessary for a 50 Ω impedance matching in air was chosen. This mismatch in air has no consequences, as the annulus is filled with materials featuring dielectric permittivities far larger than 1. A local impedance mismatch occurs at the transition of the diameter of the inner conductor. Finally, a metallic short circuit was installed at the end of the inner conductor. This leads to a sharp drop of the TDR signal at the end of the cell that can be easily identified in the TDR signal. Together with the peak of the impedance mismatch at the beginning, this can be used for a clear definition of the signal travel time. A schematic cross section of the cell is shown in Figure 2

### 2.3. TDR Device

A Sequid SDTR-65 (Sequid GmbH, Bremen, Germany) time domain reflectometer device is used for the electromagnetic measurements. The TDR device both generates a voltage step featuring a short rise time of 65 ps and records the reflected voltage signal response. In this condition, the device can operate over a large bandwidth ranging from 500 kHz to approx. 10 GHz (nevertheless, due to the limitations of used connectors, coaxial cable type and length the effective bandwidth is reduced distinctly). After a SOL (Short-Open-Load) calibration, the STDR 65 software can be calibrated in frequency domain providing the opportunity to change the rise time. Please note that the complete details, performances and features of the device can be found elsewhere [28].

In this experiment, a rise time of 1000 ns was used. This value was adjusted manually in order to fulfil a compromise. On the first hand, a sharp rise time means large frequency bandwidth. The high frequency content will decrease invariably the smoothness of the signal around impedance mismatch. The smoothness of the signal is important since we are intending to perform inversion. On the other hand, a large rise time will reduce the spatial sensitivity. Preliminary test were performed with different rise time setting. Signals obtained with a rise time of 1000 ns were considered ideal for the chosen analysis method.

### 2.4. Materials

Beads of soda-lime-glass are used as they offer several advantages for experimentally representing an idealised soil. Due to their uniformity, they are free from the influences of different angularities and density variations, while the density is still comparable to that of natural quartz based soil grains. A further benefit is a constant dielectric permittivity independent of the diameter of the particles, as no differences in mineralogy have to be taken into account. Therefore, glass beads are frequently used in erosion tests [11,29,30]. The features of the used glass beads are provided in Table 2. 

While the coarse beads (6.0 and 8.0 mm) were used as filter, two fine fractions were available as base material. The beads with 2.0 mm diameter were used as a subbase filter for the base material in order to prevent it trickling down of fine particles through the flow homogenizer. 

## 3. Spatial TDR: From TDR Trace to Porosity Profile

### 3.1. Principle

The local differences in the porosity of water saturated glass beads cause a local change in the dielectric permittivity, which affects the propagation of the electromagnetic waves along the coaxial cell and consequently alters the reflection coefficient measured at the cell input. Therefore, the measured reflected TDR signal can be analysed, aiming to determine the permittivity and thus the porosity profile along the sensor. Conventional TDR does not take advantage of this profile information as only a mean value can be obtained. Spatial TDR on the other hand uses this information [31,32]. The key feature of Spatial TDR is the inversion of the measured TDR trace in order to determine a profile of state parameters of the material under test, in this case the porosity. The inversion process of the measured signal in order to estimate the porosity profile can be decomposed into three basic steps, as schematized in Figure 3. It is based on two main assumptions: the signal propagation is non-dispersive and the spatial distribution of the porosity is unidirectional along the signal propagation in the cell. 

The total signal *U*, consisting of the incident signal *U_i_* and the reflected signal *U_r_^mes^*, relates to the transmission line parameters according to the Telegraph Equation (1)
(1)LC∂2∂t2−∂2∂x2+(LG+RC)∂∂t+∂L∂xL∂∂x+RG]U=0,
where *R*, *L*, *C* and *G* are the primary coefficients of the transmission line: *L* is the inductance, *C* the capacitance, *R* the resistance and *G* the conductance per length unit. Based on the reflection coefficient, the distribution of the discrete line parameters *C* and *G* can be reconstructed under the assumption of constant and known values for *L* and *R*. 

The capacitance is the key parameter, which we shall focus on since it depends on the dielectric properties of the material filling the coaxial cell. The resulting permittivity also depends on the geometry of the transmission line via a geometric factor. The permittivity profile can finally be obtained from the capacitance profile based on a suitable geometrical model for the cell.

In the last step, the porosity profile is computed from the permittivity profile. Different models can be applied for this last step. In this work, a physically based mixing equation was used. Details and explanations of this choice will be explained in Section 3.4.

In brief, the analysis of the TDR-signal involves the inversion of the Telegraph equations provided a suitable geometrical model and a calibrated mixing model is given. Each step of the inversion process must be adapted to the specific experimental setup. The next paragraphs introduce these steps in detail.

### 3.2. Computation of the Capacitance Profile

The cell is modelled as a transmission line with the parameters given in Figure 4. The inversion of the Telegraph equation is simplified by considering *L* and *G* as constant and *R* equal to 0. This assumption is justified as the inductance and conductance is not dependent on the permittivity of the material under test and the DC resistance is negligible for the considered condition. Therefore, only the capacitance is unknown and the profile *C*(*x*) is reconstructed using the conjugated gradient method used by [20] in his inversion algorithm. 

By discretizing the signal *U*(*x*,*t*), the Equation (2) can be solved using the Finite-Difference-Time Domain-Method (FDTD) [33].
(2)$$\left[L_{0}C\frac{\partial^{2}}{\partial t^{2}}-\frac{\partial^{2}}{\partial x^{2}}+L_{0}G_{0}\frac{\partial }{\partial t}\right]U=0,$$

The measured TDR signal *U^mes^*(*t*) is compared with the simulated signal *U^sim^*(*t*) = *U*(*x* = 0,*t*) and the error between measured and simulated signals is computed according to the cost Equation (3).
(3)$$J=\|U^{mes}-U^{sim}{\|}_{2}^{2},$$

The inversion based on the linear conjugate gradient method starts (first iteration) with an initial constant capacitance profile. At each step of the subsequent iterations, the cost function *J* is computed. At each iteration, the capacitance profile $C^{\left(k+1\right)}$ is computed according to Equation (4) as a function of the previous profile $C^{\left(k\right)}$, the cost function *J* and a factor *β* used to minimize the new cost function. This approach consists of following the opposite direction of the costs gradient, noted −∇*^C^J*, and computing an optimised step with the Nelder-Mead-Method [34] with the search by dichotomy of the factor *β*. A few iterations lead to an improved capacitance profile. It should be noted that in this method correct determination of *τ_travel_*, that is, the one way travel time along the sensor to be determined using the tangent method, is important as it fixes the mean capacitance and influences the resolution of ∇*^C^J*.
(4)$$\{\begin{array}{c} C^{\left(0\right)}=cst\\ \forall k\in N,\;C^{\left(k+1\right)}=C^{\left(k\right)}-\beta \nabla^{C}J \end{array},$$

### 3.3. Computation of the Dielectric Permittivity Profile

The second step of the inversion process is the inversion of the geometrical model defined by Equation (5).
(5)C∝ε,

For a coaxial cell, the geometric factor *g* [m] relates the capacitance *C* to the dielectric permittivity *ε* and is for the coaxial line cell a linear coefficient given by Equation (6) [35].
(6)ε=Cg with g=2πln(ba),
where *a* and *b* are the diameters of the inner and outer conductor, respectively.

### 3.4. Computation of the Porosity Profile

The third and last inversion step is the conversion of the apparent dielectric permittivity into the porosity for each discretization point. To do so, different methods can be used: empirical calibration [15], soils specific calibration for example, with other sensors [36] or mixing equation [37] (other methods such as multivariate approach [38] or numerical mixing equation [39] were not considered). Empirical calibration or specific calibration were dismissed because of the impossibility to correct the temperature effect. To properly apply these methods, the measurements have to be performed at the same temperature conditions than the calibration [23]. Considering the volume of water involved in the experiments, it was not possible to maintain a constant temperature throughout the experiment. Therefore, mixing equations were chosen in this study since they provide the opportunity to take into account temperature dependency. Such mixing equations have two disadvantages: first they usually consider a simple soil structure and second the interactions between the individual components and their contribution to the electromagnetic properties are not entirely reflected [37]. However, in the presented study, the sample can be considered fully water-saturated (without the existence of air), which simplifies the mixing equations as only two phases need to be considered, namely water and solid [40]. In this study, two types of mixing equations were tested.

The Lichtenecker-Rother model (LRM) [41] is frequently used in soil physics as mixing equation due to its simplicity. In this model, the permittivity of the mixture is the weighted sum of the dielectric properties of each individual phase multiplied by its volume fraction. In our case, the simplified LRM model is as follows:(7)$$\varepsilon ={\left(n{\left(\varepsilon_{w}\right)}^{a}+\left(1-n\right){\left(\varepsilon_{s}\right)}^{a}\right)}^{\left(1/a\right)},$$
where *ε* is the apparent dielectric permittivity of the mixture, whereas *ε_w_* and *ε_s_* are the permittivity of the liquid (water) and solid (glass beads) phase, respectively. The porosity is represented by *n* and the shape factor of the mixing model by *a*. The LRM is frequently used with a shape factor of *a* = ½ and is then called the complex refractive index model (CRIM) [42] but can also be found with a shape factor *a* = ⅓ or *a* = ⅔.

The second type of mixing equation used in the presented study is the modified self-similar Bruggeman-Hanai-Sen model (BHSM) [43]. In the case of spherical inclusion in a homogeneous matrix, this model can be expressed as:(8)$$n=\frac{\varepsilon -\varepsilon_{s}}{\varepsilon_{w}-\varepsilon_{s}}\;{\left(\frac{\varepsilon_{w}}{\varepsilon }\right)}^{b},$$

For two-phase media, the BHSM is mostly used with a shape factor of *b* = ⅓ [44]. Please note that in the two precedent equations, the relative permittivity of the beads *ε_s_* was fixed at 5.5 according to their chemical composition. The permittivity of water is computed according to [45] to take into account its temperature dependence.

To select the most efficient model, some preliminary tests were performed on perfectly known samples. To do so, a one port coaxial transmission line with a sealing system was used. The analysis used is similar to the one proposed in Reference [27]. Saturated glass beads with different sizes were characterized in terms of the complex permittivity. The spectrums were systematically analysed with different mixing equations. LRM with a shape factor equal to ⅔ was found to be the most accurate model. Please note that models were classified in terms of root mean square error (RMSE) and in terms of quality of the estimation of the porosity profile. This model was chosen within the presented study for the computation of porosity profile.

## 4. Validation of the Forward Model

### 4.1. Calibration of the Forward Model

Measurements with only water in the cell were used to calibrate the forward model. The forward model as shown in Figure 3 calculated the TDR signal for different values of *G* with *R* considered to be 0. Figure 5 shows the measured and computed signals for different values of parameter *G*.

As can be seen, the simulated and measured TDR signal corresponds well for *G* = 0.005 [S/m] and *R* = 0.0 [Ω/m]. These values are used for the upcoming investigations.

### 4.2. Validation of the Inversion

After validation of the forward model, each step of the inversion procedure has to be validated based on experiments conducted under known boundary conditions. This validation procedure is shown in Figure 6. The dielectric permittivity profile was computed using the inversion procedure from the TDR signal and in forward mode based on the known porosity profile and using the mixing equations introduced before, namely BHSM and LRM.

The cell was filled with two layers of mixtures of glass beads and water with known permittivities of the mixture in the range of 15 to 30 that are also expected during the erosion tests. Two different initial conditions for the inversion algorithm were tested. One setting (inversion 1) started with a capacitance profile derived from the actual fill of the cell (from bottom up water, flow homogenizer, glass beads-water mixture). The other setting (inversion 2) started with an average value for the capacitance over the whole cell derived from the travel time.

Both settings were tested for 5 and 20 iterations. The profiles of the dielectric permittivity inverted for the different settings are shown in Figure 7.

The inversion over more iterations leads to a sharper dielectric permittivity profile for both initial settings as can be seen at the transitions of the layers with different permittivities.

In order to examine the error associated to the inversion process, the determined permittivity profile was used as input in the forward model and the calculated TDR-signal was compared to the measured one. The principle of this validation is described in Figure 8 and the comparison of both initial settings is shown in Figure 9.

According to the obtained results (Figure 9) the used inversion approach is working successfully and with a good accuracy. The best performance, with respect to the efficiency and accuracy of the method, is achieved after 20 iterations and under known initial condition as used for inversion 1 (setting 1).

In order to identify the best mixing equation, the cell was filled with layers of glass beads of different porosities. Higher porosities result from a monodisperse packing, while lower porosities result from mixtures of beads of different diameters. The porosity of each layer was determined from the dry weight of the beads used to create a layer and the volume of the layer, which was deduced from the layer height. A deviation of the height measurement of half the size of the largest particle leads to a bandwidth of expected porosities. Care was taken to achieve fully water saturated condition.

The dielectric permittivity profile was determined based on TDR measurements using the inversion algorithm and the identified values for the parameters *G* and *R*. The resulting permittivity profiles were transformed into porosity profiles using the different mixing equations (Figure 10).

As expected LRM ⅔ present the best results in term of porosity profile computation with a RSME of 11%. Therefore, this mixing equation was used for analysing the upcoming experiments. In addition, it can be seen that the boundaries between layers of higher and lower porosities can be clearly identified. The deviation towards the water layer (at around 400 mm) can be explained by the strong alteration in permittivity at this interface, leading to partial reflections and disturbances of the signal.

## 5. Computation of Porosity Profiles during Erosion Experiments

For the ultimate test of the inversion algorithm in an erosion experiment, the cell was filled with layers of monodisperse glass beads of different sizes. From the bottom to the top, there was a subbase filter, the fine base material and a coarse filter layer on the top. Base and filter were chosen to be not geometrically stable, which means that the pores generated by the coarser beads have been larger than the size of the smaller glass beads. During the filling of the cell, the porosity of each layer was determined from the dry weight and the volume of the layer. Due to the use of monodisperse glass beads, the initial porosity distribution was relatively uniform with only slight deviations. Water temperatures during the experiments were measured in order to be able to correct the temperature dependent dielectric permittivity of water.

A water flow from the bottom to the top was introduced to induce the erosion process. After exceeding the critical hydraulic gradient, the fine base particles fluidized in the pores of the filter and filter particles moved into the based layer forming a mixing zone with a lower porosity. This process was visually monitored through the observation window. The porosity of the developing mixture zone was calculated from the alterations of the layer heights as also done by Ke and Takahashi [6], assuming that the layers that were not part of the mixing were not affected in terms of porosity changes. The porosity profile determined from the TDR measurements at different time steps of the erosion experiment are shown in Figure 11. For the sake of comparison, Figure 11 also provides the porosities computed from the layer heights as grey columns and also shows a picture of the sample taken through the window. 

As can be seen from the comparison in Figure 11, a good correlation between the computed and given porosity profiles can be seen with values for the RMSE varying between 2.3% and 5.8%. After the onset of the erosion process, the transition between the layers show distinct changes in the profile. Thicker layers were identified more clearly. The reason is that the resolution of the porosity measurements with TDR is limited to the centimetre-range due to the rise time of the signal. Thin layers may not be fully recovered. Sudden jumps in the permittivity lead to part-reflections of the signal at the transitions as visible in the graph on peaks and corrugations. 

Apart from these minor issues, the proposed electromagnetic measurement technique, which combines the use of a coaxial cell and a forward transmission line model combined with an appropriate inversion approach, has shown its ability to determine porosity profiles not only for stationary states but also for relatively fast changing porosity changes in erosion tests.

## 6. Conclusions

This contribution describes the design and the validation of an experimental setup with a large coaxial erosion cell and introduces the principles of a Spatial TDR measurement approach involving a transmission line model with the aim to determine a porosity profile throughout an erosion experiment.

The cell was designed to be a sensor by itself consisting of a rigid coaxial transmission line. The soil sample is arranged in the annulus between inner and outer conductor and the erosion process of the sample inside the cell is triggered by a water flow against the direction of gravity. The hydraulic boundary conditions are monitored by pressure transducers along the cell and a flow meter. The dimensions of the setup is unique and the concept of monitoring a transient process with changing porosity conditions in a coaxial cell is original.

A key for the efficiency of Spatial TDR is the inversion principle, which consists of three fundamental steps explained in detail within this contribution. The first step is the computation of the capacitance profile along the cell based on TDR measurements. This step is computational expensive and requires some time. The second step is the calculation of the dielectric permittivity profile along the cell from the capacitance profile using a geometrical function specific to the cell. Finally, the porosity profile is computed from the dielectric permittivity profile using a calibrated mixing equation.

Every step of the proposed method and the used models have been validated individually. Measurements with media of known dielectric permittivity have been used to validate the inversion algorithm. The inversion procedure was designed to be fast, robust and efficient. Erosion tests show a good accuracy with RMSE values varying between 2.3% and 5.8% with a satisfactorily high resolution of the determined porosity profiles. 

Future erosion experiments will focus on the evolution of the mixing zone at the interface between base and filter material to improve our understanding of the onset and progress of internal erosion.

## Figures and Tables

**Figure 1 sensors-19-00611-f001:**
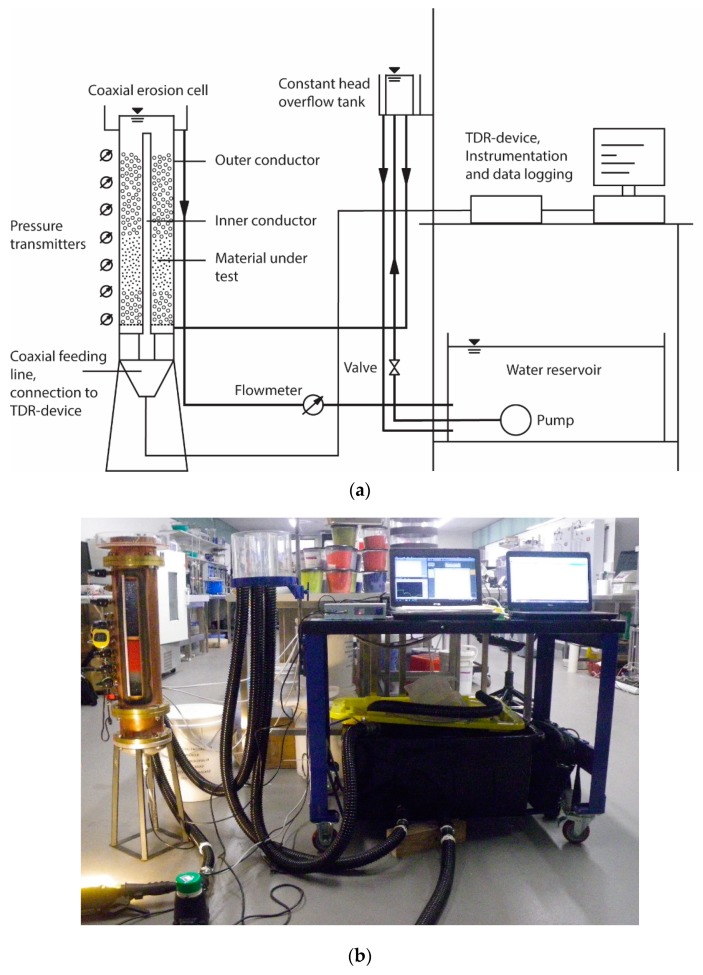
(**a**) Scheme of the experimental setup with the coaxial erosion cell (not to scale); (**b**) photograph of the setup.

**Figure 2 sensors-19-00611-f002:**
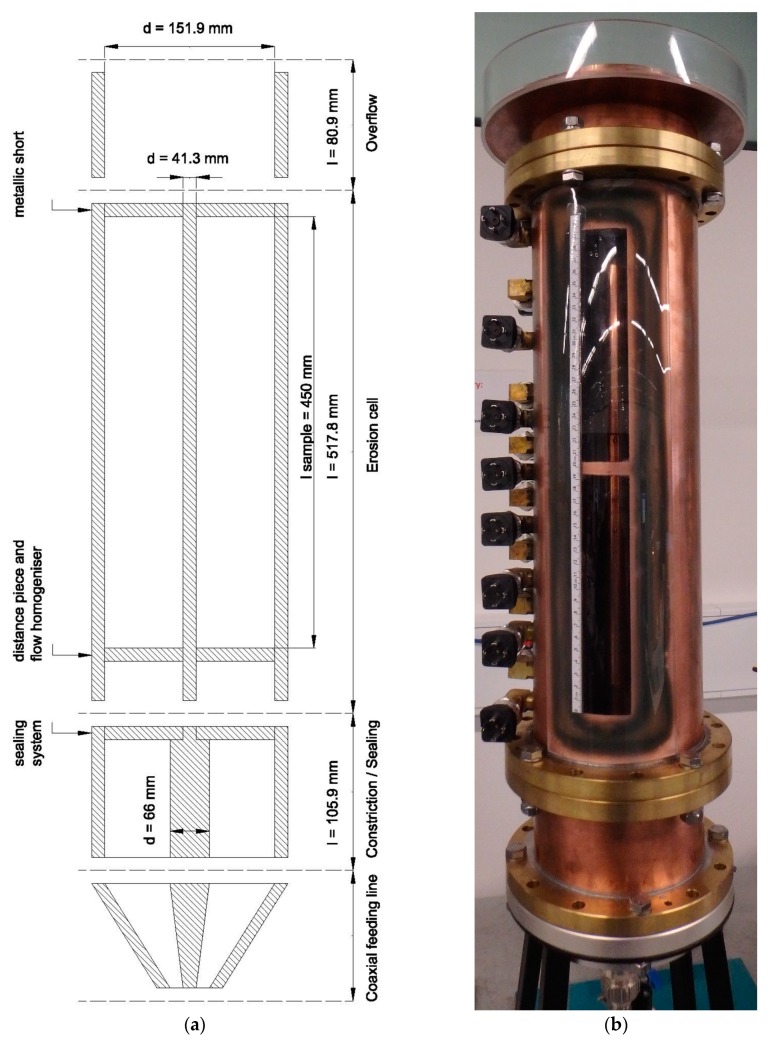
(**a**) Schematic of the coaxial erosion cell (not to scale); (**b**) photograph of the cell on the right: Pressure transducers are visible on the left hand side of the photograph and the inner conductor through the inspection window.

**Figure 3 sensors-19-00611-f003:**
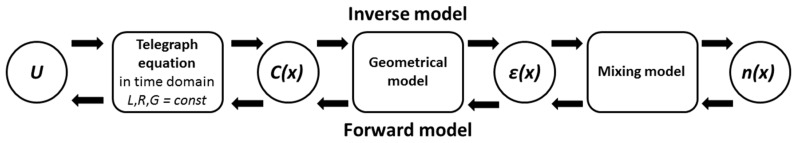
Schematic diagram of the generic inversion method to obtain the porosity profile from the measured TDR signal at the cell input.

**Figure 4 sensors-19-00611-f004:**
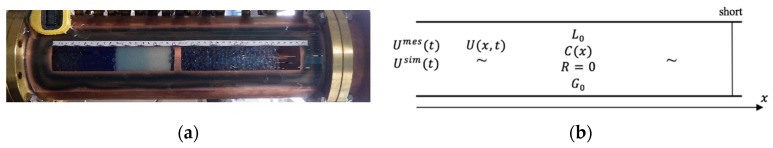
(**a**) Picture of the coaxial cell and (**b**) corresponding model of the transmission line.

**Figure 5 sensors-19-00611-f005:**
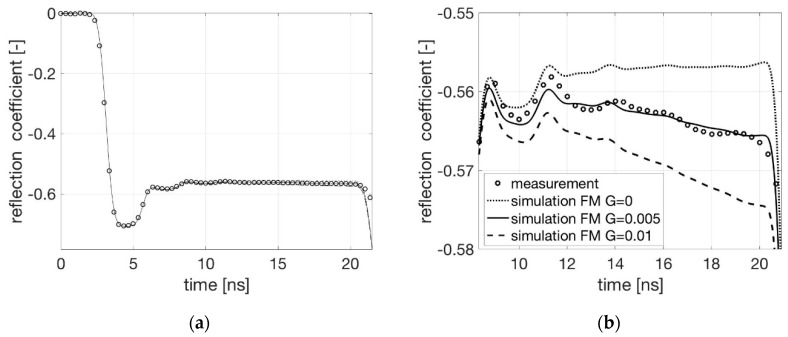
Comparison between measured and modelled TDR signals for only water in the cell and for different *G*-values with *R* = 0. (**a**) Entire signal; (**b**) An enlarged section.

**Figure 6 sensors-19-00611-f006:**
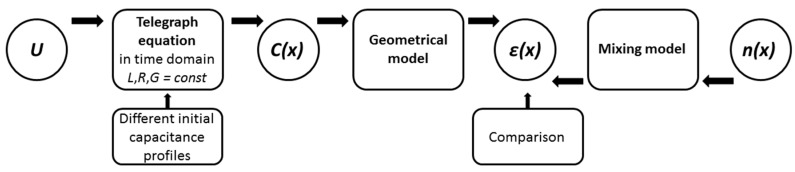
Scheme of the validation of the inversion algorithm.

**Figure 7 sensors-19-00611-f007:**
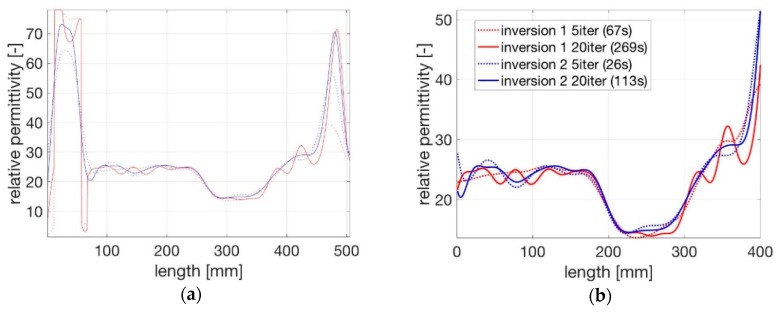
Profiles of dielectric permittivity for two settings after 5 and 20 iteration steps. (**a**) Whole signal along the longitudinal axis of the cell; (**b**) An enlarged section covering the actual soil sample.

**Figure 8 sensors-19-00611-f008:**
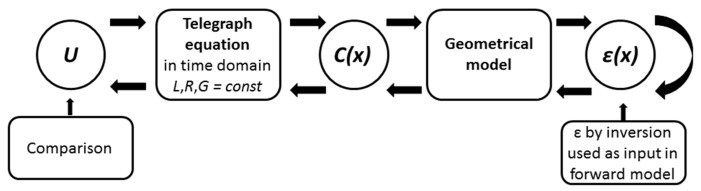
Principle of the examination of the error made in the inversion process.

**Figure 9 sensors-19-00611-f009:**
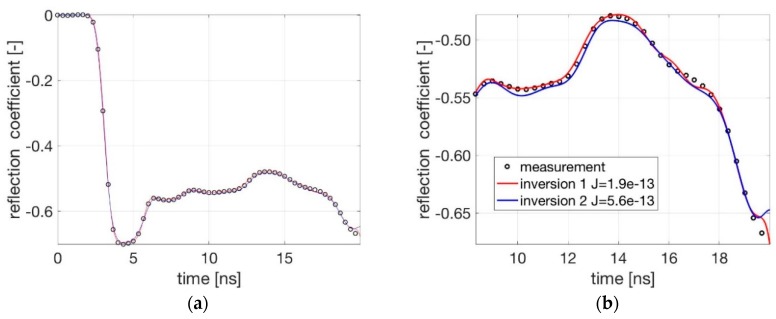
Comparison between measured and inverted reflected signals. (**a**) Whole signal; (**b**) Enlarged section of the signal sample corresponding to the phenomena occurring along the sample.

**Figure 10 sensors-19-00611-f010:**
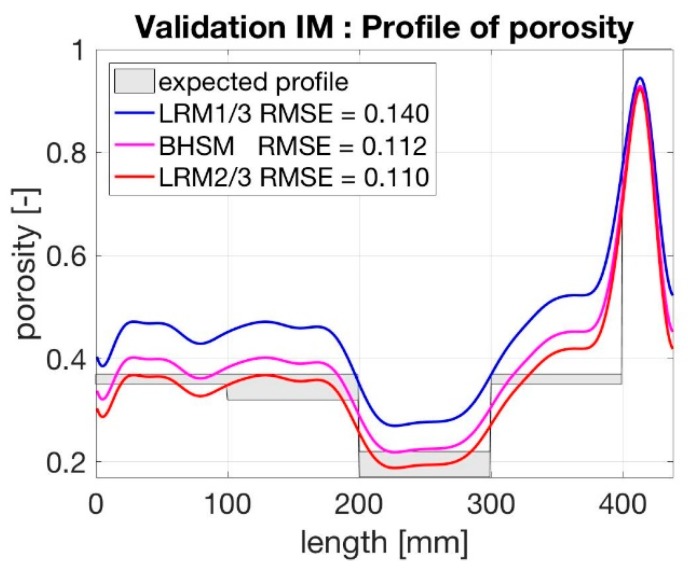
Porosity profiles calculated with the two mixing equations and two different values for the shape factor *a* in the LRM in comparison to the given porosity distribution shown as grey columns taking into account possible deviation in the height measurement.

**Figure 11 sensors-19-00611-f011:**
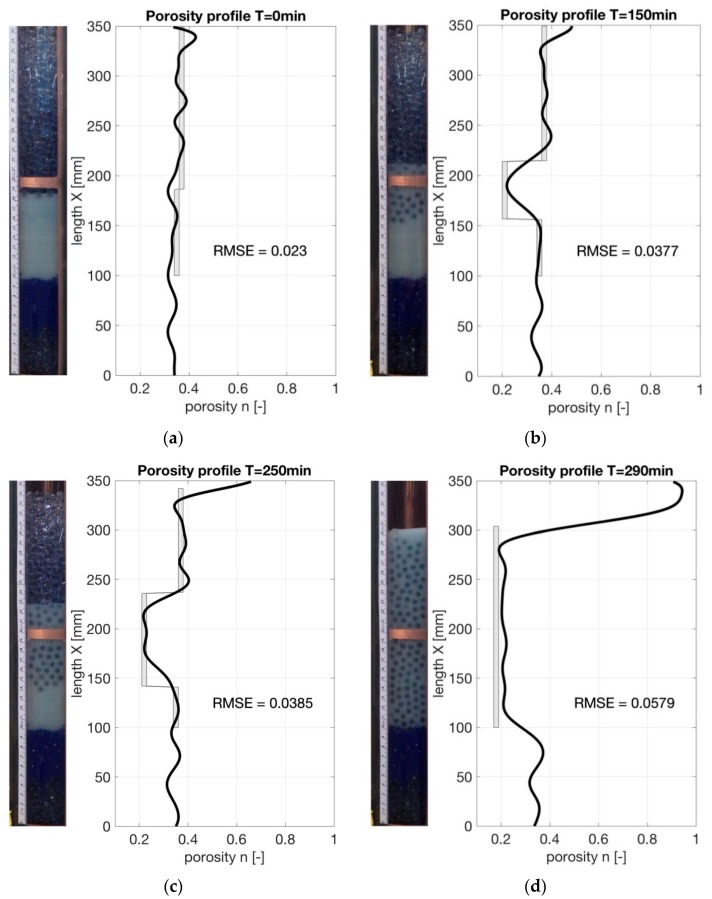
Comparison of porosity profiles determined with a TDR approach (black) and computed from layer heights (grey) at different time steps of the erosion experiment: (**a**) initial state; (**b**) after 150 min; (**c**) after 250 min, (**d**) after 290 min at the end of the test run.

**Table 1 sensors-19-00611-t001:** Features of the tubes of the coaxial erosion cell according to the data sheet of the manufacturer.

Application	Material	Diameter Inner/Outer	Spinner Component
Outer tube	Copper	151.9/155.6 mm	6 1/8” EIA outer conductor
Inner tube before constriction	Copper	64.0/66.0 mm	6 1/8” EIA inner conductor
Inner tube after constriction	Copper	38.8/41.3 mm	1 5/8” EIA outer conductor

**Table 2 sensors-19-00611-t002:** Properties of the glass beads used in the experiments.

Diameter	Roundness	Colour	Grain Density	Application
0.3–0.425 mm	≥70%	Clear	25 kN/m^3^	Base II
0.425–0.6 mm	≥70%	Red	25 kN/m^3^	Base I
2.0 mm	90%	Clear	25 kN/m^3^	Subbase filter
6.0 mm	≥90%	Clear	25 kN/m^3^	Filter D
8.0 mm	≥90%	Clear	25 kN/m^3^	Filter A

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
