# Peer review of "Determination of the Porosity Distribution during an Erosion Test Using a Coaxial Line Cell"

_sensors, 2019, doi:10.3390/s19030611_

Round 1

Reviewer 1 Report

22-23 reword sentence

37 on not of

40 reword, this sentence is unclear, presumably meaning the following approaches have been reported in the literature

41 what is the rating of the fine fraction?

64 use 1970s

77-78 respect? reword whole sentence

81-82 use cohesiveness, viscosity

88 e.g. not i.e.

100, 102 remove firstly and secondly, confusing with subsequent first and second.

116 32 mm pipes do not avoid head loss, maybe make it negligible

118/119 this sentence is irrelevant, delete

116 doesn't avoid head loss, makes them negligible

132 Avoid phrases such as 'Thanks to'

133 reword senstence

140/141 reword sentence

153 'thanks to'

177 explain why dispersion and issue, and is it travel time or amplitude change that is the offending aspect, can you quantify the effect? Why 1ns risetime ?

192 I don't think flow homogeniser had been introduced here, explain

227 and later on in manuscript. Here empirical models are dismissed as insufficiently accurate (not quantified in any way) and later several models were used and the 'best fit' adopted. This is a weak approach without justification or explanation about what is likely to be going on and with a relatively simple substrate as glass beads too.

236 is not

237 what is the condition for dc resistance to be 0? What does that mean for G?

Eqtn 3 Why is the square of the L2 norm used?  why not just the L2 norm itself?

Eqn 4 How is beta calculated? Polak-Ribiere formulation?

294 you stated earlier that G was zero, or at least only C was used in the inversion process.

Fig 7 for (b), use same terminology when referring to view with expanded axes scaling

340 What is the justification for selecting this model - just because the fit was better using data from this one experiment?

362 and Fig 11. Does the window allow much of depth into the cell to be seen or only of the beads immediately adjacent to the window - if the latter, one would expect these to be distributed differently due to the smaller flow velocity near the walls.

397 'at last' is an unsuitable turn of phrase. 

Author Response

Dear Reviewer,

Thank you for your comments on this paper. They will for sure help to improve the language and the presentation of our research.

In the following, you will find the replies to each of your specific comments or suggestions:

Line 22-23: reword sentence

Answer: Sentence is rewritten

Line 37 on not of

Answer: Thanks for your correction

Line 40 reword, this sentence is unclear, presumably meaning the following approaches have been reported in the literature

Answer: You are right. The sentence has been rewritten for clarification

Line 41 what is the rating of the fine fraction?

Answer: The fine fraction was rated by weight of the washed out fine fraction over the time or time periods of their experiments by build in scales. After the experiments, a sieve analysis of the washed out fine fraction was carried out in order correlate mass, grains size distribution and hydraulic boundary conditions. In some experiments, this was accompanied by layer-wise sieve analysis of the remaining coarse soil skeleton in order to locate the origin of the washed out fraction.

This is now better explained in the paper.

Line 64 use 1970s

Answer: Changed.

Line 77-78 respect? reword whole sentence

Answer: Sentence is rewritten.

Line 81-82 use cohesiveness, viscosity

Answer: Expressions changed in the paper.

Line 88 e.g. not i.e.

Answer: Corrected in the paper

Line 100, 102 remove firstly and secondly, confusing with subsequent first and second.

Answer: Concerned sentences are rewritten.

Line 116 32 mm pipes do not avoid head loss, maybe make it negligible

Answer: This is true and clarified in the paper. For the flow velocities in the pipes for the experiments presented here, no head loss was detected with this diameters leading to this unprecise wording.

Line 118/119 this sentence is irrelevant, delete

Answer: Sentence removed.

Lined 116 doesn't avoid head loss, makes them negligible

Answer: True, answer see above.

Line 132 Avoid phrases such as 'Thanks to'

Answer: Concerned sentences rewritten.

Line 133 reword senstence

Answer: Sentence rewritten for clarification.

Line 140/141 reword sentence

Answer: Sentence rewritten for clarification.

Line 153 'thanks to'

Answer: Concerned sentences rewritten.

Line 177 explain why dispersion and issue, and is it travel time or amplitude change that is the offending aspect, can you quantify the effect? Why 1ns risetime ?

Paragraph 2.3 changed in the paper for better explanation.

Line 192 I don't think flow homogeniser had been introduced here, explain

Answer: The flow homogeniser was introduced in line 153. It is a perforated plate of PMMA. It is located between the water-inlet of the cell and carries the soil column (or in this case the glass beads). Due to this part, the hydraulic head is equal at the bottom of the soil column leading to more uniform flow conditions in the pores. Tests without this part have shown a non-uniform progress of erosion, visible through different settlements of the soil.

This part is now described more in detail for clarification.

Line 227 and later on in manuscript. Here empirical models are dismissed as insufficiently accurate (not quantified in any way) and later several models were used and the 'best fit' adopted. This is a weak approach without justification or explanation about what is likely to be going on and with a relatively simple substrate as glass beads too.

Combined answer for comments/questions regarding line 227 and 340:

Firstly, specific empirical calibration was dismissed because of the problem because of temperature correction. Indeed, it is important to keep in mind that the erosion experiment is made with a large volume of water (several litter) in laboratory conditions in Australia during summer. Thus, it was not possible to maintain a constant temperature of the water: specific empirical calibration would have bring extra errors. This is why we selected mixing equation. (Please note that the effect of temperature on calibration was investigated in [U1]).

Mixing equations are numerous, thus the selection of the proper model is fundamental. To do so, some preliminary measurements were performed. Please note that we decided not to include these tests in the paper in order to make it more clear (introducing these preliminary measurements would have made the paper very long and not easy to read !).  This is why we used the simple and pragmatic argument of the “best fit” which is in adequacy with the preliminary results.

Nevertheless, here are some details about the preliminary test:

In these experiments a coaxial transmission line used only in reflection was used. Saturated glass beads was inserted in the line (measurement with different size of beads were performed). From the measured scattering function, the complex permittivity was determined (the analysis used here is similar to the one proposed in [U2]). Then, different 2 phases mixing equations were tested. In this configuration, the porosity of the sample could be determined by gravimetric measurement: the measured value was used as an input in the mixing equation. Finally, several model were tested: BHSM, LRM 1/3, CRIM (or LRM ½), LRM 2/3 and systematically compared in term of RMSE. The results have shown that LRM 2/3 was the best model

[U1] Scheuermann, A. (2012). Determination of porosity distributions of water saturated granular media using spatial time domain reflectometry (spatial TDR). Geotechnical Testing Journal, 35(3), 441-450

[U2] Bore, T., Bhuyan, H., Bittner, T., Murgan, V., Wagner, N., & Scheuermann, A. (2017). A large coaxial reflection cell for broadband dielectric characterization of coarse-grained materials. Measurement Science and Technology, 29(1), 015602.

Paragraph beginning at line 227 rewritten

Paragraph 3.4 partly rewritten.

Line 236 is not

Answer: Sentence rewritten.

Line 237 what is the condition for dc resistance to be 0? What does that mean for G?

Answer: R, proportional to the resistivity  , is equal to 0 because no ohmic losses were assumed for the cell.

G is proportional to the conductivity . With the use of water ( ) it cannot be assumed that G is equal to 0. However, the authors did the hypothesis that G is almost constant during the tests all along the cell.

Eqtn 3 Why is the square of the L2 norm used?  why not just the L2 norm itself?

Answer: This was done with the purpose to give more weight/influence to values which are not reached during the process of inversion. This approach was suggested by S. Schlaeger. A fast TDR-inversion technique for the reconstruction of spatial soil moisturecontent. Hydrology and Earth System Sciences Discussions, European Geosciences Union, 2005, 9 (5), pp.481-492.

Eqn 4 How is beta calculated? Polak-Ribiere formulation?

Answer: For these tests the linear conjugate gradient method was used. The direction   is equal to  . b is step optimized, computed by Nelder-Mead method.

Indeed, it can be possible to use the nonlinear conjugate gradient method giving a direction   computed by Fletcher-Reeves-Koeffizienten (FRK) or Polak-Ribiere-Polyak (PRP) methods.

Line 294 you stated earlier that G was zero, or at least only C was used in the inversion process.

Answer: We stated that G is constant and only C is dependent on the permittivity profile.

Fig 7 for (b), use same terminology when referring to view with expanded axes scaling

Answer: Figure captions rewritten

Line 340 What is the justification for selecting this model - just because the fit was better using data from this one experiment?

Answer: As expected LRM 2/3 present the best results in term of porosity profile computation. The other models were checked for verification of this expectation. The RSME for the tested mixing models is added.

Line 362 and Fig 11. Does the window allow much of depth into the cell to be seen or only of the beads immediately adjacent to the window - if the latter, one would expect these to be distributed differently due to the smaller flow velocity near the walls.

Answer: The authors also were at first unsure if the relatively small window would allow the progress of the erosion process accurately enough. However, preliminary tests without TDR-measurements in a cell made of PMMA (thus all walls are transparent) showed that this concerns were unnecessary as the erosion progress was very evenly. The observations during the experiments have shown that the progress in the coaxial cell was very evenly as well.

Because of the clear glass beads with a relatively large diameter as filter material, a depth of at least three times the filter was visible through the window showing an even progress. At the last stage of the tests when the base material reached the top of the filter column (compare Figure 11 d), an even progress was also visible at the surface of the column.

This is the case for the monodisperse glass beads used. For graded mixtures and real soils materials, a more uneven progress has to be expected.

Line 397 'at last' is an unsuitable turn of phrase.

Answer: The sentence is rewritten.

Reviewer 2 Report

The research contribution was quite interesting although very much out of my own research scope. This is quite a technical paper that I am sure will appeal to the target audience of soil physicians and engineers. What i did not get a sense of in this paper was how this technology could be applied and what contexts it could operate in. Probably would like to see a bit more higher level discussion of where this would fit into for example agricultural context such as improving decisions around irrigation management, or how it could improve soil testing and what further information we can retrieve about soils.  

As a soil scientist, internal erosion is not a term I am familiar with. It is probably an engineering term. With what is described, the process appears to be called lessivage (https://www.sciencedirect.com/science/article/pii/S0016706111002345). Maybe to revise, authors need to acknowledge this.

Typo on line 68

Line 77: “One of them is the respect of the representative elementary volume criteria (REV)” . Grammar does not seem right here.

Typo line 395

Author Response

Dear Reviewer,

Thank you for your comments on this paper. They will for sure help to improve the presentation of our research.

In the following, you will find the replies to each of your specific comments or suggestions:

Comment:

The research contribution was quite interesting although very much out of my own research scope. This is quite a technical paper that I am sure will appeal to the target audience of soil physicians and engineers. What i did not get a sense of in this paper was how this technology could be applied and what contexts it could operate in. Probably would like to see a bit more higher level discussion of where this would fit into for example agricultural context such as improving decisions around irrigation management, or how it could improve soil testing and what further information we can retrieve about soils.  

As a soil scientist, internal erosion is not a term I am familiar with. It is probably an engineering term. With what is described, the process appears to be called lessivage (https://www.sciencedirect.com/science/article/pii/S0016706111002345). Maybe to revise, authors need to acknowledge this.

Answer:

Internal erosion is indeed a term in geotechnical engineering. It is the transport of granular particles induced by the force of flowing water. This can occur at an interface between a coarse and fine soil as in the paper (contact erosion) or for example with mobile fine particles in the skeleton of a coarse fraction, which is called suffusion. Internal erosion is a physical process, which can occur in very short time.

As the authors are no soil scientists, lessivage is not a familiar process. However, from the paper mentioned by the Reviewer, it appears that lessivage is more a long-term process of cohesive soils with also a chemical (pH) aspect and may thus to the authors opinion not be directly comparable.

Regarding the use and application of this cell: It is intended mostly for laboratory experiments on internal erosion, as the mechanisms that induce and maintain this erosion process are still partly unknown. The determination of porosity changes in this process is an important part towards better understanding. The glass beads used in the paper were necessary for validation, erosion test with granular soils are also possible.

Apart from this, the technique is suitable for processes in which the dielectric permittivity undergoes changes. In the context of soils, this is the volumetric water content in unsaturated conditions or the porosity in saturated conditions.

For agricultural applications, TDR rod-probes exist and are available commercially which can determine the local water content.

Typo on line 68

Answer: Noted.

Line 77: “One of them is the respect of the representative elementary volume criteria (REV)” . Grammar does not seem right here.

Answer: The sentence is rewritten.

Typo line 395

Answer: Noted.

Reviewer 3 Report

This article concerns the detection of porosity changes within a soil matrix caused by internal erosion using Spatial Time Domain Reflectometry (STDR) and a transmission line model is proposed for this purpose.

The paper is well written. This paper does a good job of “introducing” a new method for evaluation the soil porosity from the dielectric permittivity profile by physically based mixing rules.

Specific comments

In the abstract a number of quantitative statistical indexes related to the results of this study should be added.

l.40. You should revised the word “haven”

l. 173-179. What is the radius of effect of the electromagnetic wave?

l.204. Give a brief outline of the basic operating principle of the device SDTR.

l.340 How different are the results compared to the modified self-similar Bruggeman Hanai Sen model (BHSM) and the LRM model with a shape factor of α = ½. Please provide a statistical index e.g RMSE

Author Response

Dear Reviewer,

Thank you for your kind comments on this paper. They will for sure help to improve the presentation of our research.

In the following, you will find the replies to each of your specific comments or suggestions:

Comment:

In the abstract a number of quantitative statistical indexes related to the results of this study should be added.

Answer:

The RSME-values are added in the abstract and later on in section 4.1 and 5.

Line 40. You should revised the word “haven”

Answer: Wrong spelling corrected.

Line 173-179. What is the radius of effect of the electromagnetic wave?

Answer: The sensitive area, in other words the area covered by the electromagnetic field is the annulus between the inner and outer conductor. The diameter of the inner conductor is 38.8 / 41.3 (inner / outer) and that of the outer conductor is 151.9 / 155.6 (inner / outer).

Line 204. Give a brief outline of the basic operating principle of the device SDTR.

Answer: Please see the modified section 2.3 TDR-device.

Line 340 How different are the results compared to the modified self-similar Bruggeman Hanai Sen model (BHSM) and the LRM model with a shape factor of α = ½. Please provide a statistical index e.g RMSE

Answer: The RSME is provided for the comparison of the used mixing-models and added in Figure 10.

Round 2

Reviewer 1 Report

40 In my first review: Line 37 on not of Answer: Thanks for your correction

Could you please clarify further, I may have misinterpreted what you meant. I think you meant that "... and are dependent on the mechanical and hydraulic properties." not "... and are independent of the mechanical and hydraulic properties."

245  "Details and explanations of this choice will be explained later."  Specify the Section number rather than saying "later".

315 "LRM with a shape factor equal to 2/3 was found to be the more robust model." There are various ways of defining robustness. Specify what parameter(s) you used to select  LRM with a shape factor equal to 2/3.

Eqn 3 refer to question from previous review and your response.  define how beta is calculated re provide reference - you answered in reviewer response but did not correct. Also, there is another "thanks to" - delete it.

324 for G=0.005 not 0.05 ?

Author Response

Dear Reviewer,

thank you for your comments. The authors hope that now the last points and questions are clarified.

Comment:

40 In my first review: Line 37 on not of Answer: Thanks for your correction

Could you please clarify further, I may have misinterpreted what you meant. I think you meant that "... and are dependent on the mechanical and hydraulic properties." not "... and are independent of the mechanical and hydraulic properties."

Answer:

The expression interdependent (not independent) was chosen to highlight that the micro-scale properties (pore shapes) and the macro-scale properties (porosity, mechanical and hydraulic properties) are interconnected to each other.

Comment:

245  "Details and explanations of this choice will be explained later."  Specify the Section number rather than saying "later".

Answer:

It is meant section 3.4. This is specified in the paper.

Comment:

315 "LRM with a shape factor equal to 2/3 was found to be the more robust model." There are various ways of defining robustness. Specify what parameter(s) you used to select  LRM with a shape factor equal to 2/3.

Answer:

The expression “robust” can indeed be misleading here. It was changed to a more appropriate word and the paragraph was rewritten as follows:

“LRM with a shape factor equal to 2/3 was found to be the most accurate model. Please note that models were classified in terms of root mean square error (RMSE) and in terms of quality of the estimation of the porosity profile.”

Comment:

Eqn 3 refer to question from previous review and your response.  define how beta is calculated re provide reference - you answered in reviewer response but did not correct. Also, there is another "thanks to" - delete it.

Answer:

As beta is mentioned, it seems that this question is related to Eqn. 4. This is now clarified in the paper in the lines 269 to 272.

Comment:

324 for G=0.005 not 0.05 ?

Answer:

G = 0.005 S/m (conductivity of drinking water) is the correct value. Corrected in the paper.
